# Characteristics of chemically induced liver progenitors derived from a pig model of metabolic dysfunction-associated steatotic liver disease

**Masayuki Fukumoto**[1], **Daisuke Miyamoto**[1], **Akihiko Soyama**[1], **Takanobu Hara**[1], **Yasuhiro Maruya**[1], **Peilin Li**[1], **Hajime Matsushima**[1], **Kazushige Migita**[1], **Takahiro Enjoji**[1], **Hanako Tetsuo**[1], **Takuro Fujita**[1], **Mampei Yamashita**[1], **Hajime Imamura**[1], **Tomohiko Adachi**[1], **Kengo Kanetaka**[1], **Takahiro Ochiya**[2], **Susumu Eguchi**[1]*

**1** Department of Surgery, Nagasaki University Graduate School of Biomedical Sciences, Nagasaki, Japan,
**2** Department of Molecular and Cellular Medicine, Tokyo Medical University, Tokyo, Japan

* sueguchi@nagasaki-u.ac.jp

**Data Availability Statement:** All relevant data are within the manuscript and its Supporting Information files.

## Abstract

We previously reported the efficacy of chemically induced liver progenitors (CLiP) as a source of cells for transplantation in patients with liver disease. This study aimed to characterize CLiP derived from steatotic livers using a pig model for future clinical applications. Livers were removed from miniature pigs with diet-induced steatosis and normal livers by laparoscopic hepatectomy. Mature hepatocytes (MH) isolated from the livers of each group were cultured in differentiation medium composed of Y-27632, A-83-01, and CHIR99021 (YAC medium). The characteristics of CLiP, including liver-specific function, proliferative capacity *in vivo*, and extracellular vesicles (EVs) production, were evaluated. Although CLiP in both groups expressed hepatic progenitor cell markers (Epithelial cell adhesion molecule and Trophoblast cell surface antigen 2), the proliferative potential was higher for the disease group than the healthy group. In contrast, markers of functional MH after re-differentiation were only detected in the healthy group. Both groups showed high cell viability and the ability to differentiate into albumin-positive cells *in vivo*. EVs counts were lower in disease-derived CLiP than in the normal group; however, there were no differences in microRNA expression within EVs. Using a pig model, CLiP was successfully produced from a liver that reproduced steatotic liver disease. Although there were slightly fewer EVs from CLiP in the disease group than in the normal liver group, the *in vivo* proliferative capacity of CLiP was high. Therefore, CLiP induced in the steatotic liver are a promising source for cell therapy in patients with liver disease.

## Introduction

Liver transplantation is a life-saving treatment for patients with end-stage liver disease [1,2]. Among the various indications for liver transplantation, metabolic-dysfunction-associated

**Funding:** This study was supported in part by the Japan Agency for Medical Research and Development (Grant No. JP22bk0104152). The funders had no role in study design, data collection and analysis, decision to publish, or preparation of the manuscript.

**Competing interests:** The authors have declared that no competing interests exist.

steatotic liver disease (MASLD) is becoming more prevalent and may surpass all other indications shortly [2]. However, liver transplantation is limited by the persistent mismatch between the number of patients on the waiting list and organ donations as well as the burden on living donors. Therefore, liver regenerative medicine is a promising alternative therapy [3,4], including cell transplantation therapy with the administration of single cells [5,6]. Hepatocyte transplantation has been evaluated in numerous clinical trials, mostly with primary hepatocytes [7,8]. However, the availability of primary mature hepatocytes (MH) is limited. MH that cannot proliferate in vitro are a limiting factor in the clinical application of hepatocyte transplantation.

Liver progenitor cells are a promising source for liver disease therapy owing to their proliferative abilities and bidirectional differentiation potential. Cai et al. reported that liver progenitor cell transplantation improves liver function [9] and Takeuchi et al. reported that extracellular vesicles (EVs) from transplanted mesenchymal stem cells can treat liver fibrosis by activating stellate cells [10]. Furthermore, it has been reported that EVs contribute to the inhibition of fibrosis [11]. Therefore, cell transplantation using liver progenitor cells is expected to be an effective treatment for diseases, such as liver cirrhosis.

We have previously described chemically induced liver progenitors (CLiP), which is liver progenitor cell induced by stimulation with small molecules, including Y-27632 (Rho-associated, coiled-coil containing protein kinase (ROCK) inhibitor), A-83-01 (inhibitor of TGF-b type I receptor), and CHIR99021 (GSK3b inhibitor) [12–14]. CLiP differentiate into MH and biliary epithelial cells in vitro and have antifibrotic effects when transplanted into liver disease models [13]. We chose a diet-induced steatohepatitis rat model as a liver disease model; this is to represent MASLD, which is currently recognized as the most common form of chronic liver disease in developed countries [13].

Furthermore, we have successfully isolated hepatocytes from livers harvested from patients with cirrhosis and converted these hepatocytes into CLiP for applications in autologous transplantation therapy in clinical practice [14]. However, it is not clear whether the characteristics of CLiP, including their function, proliferative capacity, and

EVs production, differ among different liver backgrounds.

The aim of this study was to clarify the characteristics (e.g., cell proliferative capacity, responsiveness, and functions) of porcine CLiP (pCLiP) induced from MH isolated from the liver of a miniature pig model of diet-induced steatosis with fibrosis, resembling MASLD. These results have potential clinical applications for autologous transplantation of CLiP induced from resected livers in humans.

## Materials and methods

### Hepatocyte isolation from pigs

All pigs were treated according to the Guidelines of the Intervention Technical Center Kobe laboratory (IVTeC Kobe lab) on Animal Use with the protocol approved by the Ethics Committee for Animal experimentation of IVTeC Kobe lab (approval number, IVT23-28). Liver tissues were removed from healthy and diseased clawn minipigs (female, 25–35 kg; Kyoto Animal Inspection Center Co.) using a laparoscopic technique. Approximately 30 g of each tissue was removed at the IVTeC facility (Kobe, Japan). A mixture of Ketamine (10 mg/kg) and Xylazine (2 mg/kg) was administered intramuscularly to sedate the pigs. For induction of anesthesia, after the inhalation of a high concentration of Isoflurane, a tracheal tube was inserted. The pigs were anesthetized with isoflurane during the partial liver resection with intravenous administration of lactate Ringer's solution.

The protocol was set by assuming the future clinical application of cell transplantation of CLiP induced from a partially resected liver by a laparoscopic procedure. Isolated pig liver tissue was de-blooded with Hank's Balanced Salt Solution (HBSS), washed with University of Wisconsin Solution (UW), and then transported to Nagasaki University under cold storage conditions (reflecting conditions for potential clinical applications). The duration of liver transport was approximately 5 h. Porcine mature hepatocytes (pMH) were isolated from collected tissues using a modified two-step collagenase perfusion method [14]. The isolated cells were filtered through a cotton mesh membrane and 63-μm stainless mesh and then purified thrice via centrifugation at $50 \times g$ for 2 min each at 4˚C. The cell suspension was subsequently mixed with 25% Percoll Plus solution (GE Healthcare, Tokyo, Japan) via centrifugation at $70 \times g$ for 7 min at 4˚C.

## Hematoxylin and eosin (H&E), silver, and desmin staining

A portion of the excised liver was fixed in 10% formalin neutral buffer (Kenei-Pharm, Osaka, Japan). Fixed samples were embedded in paraffin, cut into 5 μm cross-sections, and mounted on Matsunami CREST-coated slides (CRE-05; Matsunami Glass, Osaka, Japan). The sections were deparaffinized and stained with Mayer's Hemalum Solution (1:1; 3000–2; Muto Pure Chemical, Tokyo, Japan), Hematoxylin and Eosin (H&E) (1:3 with 95% alcohol, 3204–2; Muto Pure Chemical), and silver (19511; Muto Pure Chemical). For immunostaining, sections were heated in citrate buffer (pH 6.0) using a microwave at 99˚C for 20 min for antigen retrieval and incubated in 0.3% hydrogen peroxide methanol (25183–81 and 18084–01; Kanto Chemical, Tokyo, Japan) for 20 min to quench endogenous peroxidase activity. Sections were incubated overnight at 4˚C in Tris-buffered saline (TBS) with the following antibodies: rabbit anti-desmin (1:200, 41291; Nichirei Biosciences, Tokyo, Japan). The sections were visualized using a Simple Staining System (MAX-PO, Nichirei, Japan) with a reaction time of 30 min. The red reaction was performed using ImmPACT NOVA RED (SK-4805; Vector Labs, Newark, CA, USA). Bright-field images were captured using an optical microscope (Axio Imager.A1; ZEISS, Oberkochen, Germany).

## Chemical reprograming to pCLiP

Porcine MH were seeded onto collagen type I-coated dishes (Asahi Techno Glass, Tokyo, Japan) at a density of $2 \times 10^4$ cells/cm$^2$ in STIM to promote attachment to the plate surface. The STIM medium was obtained from a Hepatocyte Culture Media Kit with 10 ng/μL epidermal growth factor (EGF) containing 1× penicillin-streptomycin-glutamine (100×) (Gibco, Waltham, MA, USA) and 10% fetal bovine serum (FBS; Gibco). After 1 d of culture, the culture medium was changed to a small, chemically reprogrammed culture medium including Dulbecco's Modified Eagle Medium (DMEM)/F12 containing 2.4 g/L NaHCO$_3$ and L-glutamine (Life Technologies) supplemented with 5 mM 4-(2-hydroxyethyl)-1-piperazineethanesulfonic acid (HEPES), 30 mg/L L-proline, 0.05% bovine serum albumin (BSA), 10 ng/mL EGF (all from Sigma-Aldrich Japan, Tokyo, Japan), insulin-transferrin-serine (ITS)-X (Life Technologies), $10^{-7}$ M dexamethasone (Dex) (Fuji Pharma Co. Ltd., Tokyo, Japan), 10 mM nicotinamide (Sigma-Aldrich Japan), 1 mM ascorbic acid-2 phosphate (Wako Pure Chemical), 100 U/mL penicillin, and 100 mg/mL streptomycin (Life Technologies) in addition to 10 μM Y-27632 (AdooQ BioScience, Irvine, CA, USA), 0.5 μM A-83-01 (Wako Pure Chemical), 3 μM CHIR99021 (AdooQ BioScience), and 20 ng/mL recombinant hepatocyte growth factor (rHGF). The culture medium was changed every 2 or 3 days. It took 14–16 days for porcine CLiP (pCLiP) to reach 90% conference from the pMH.

### Immunofluorescence staining

pCLiP were fixed with 4% paraformaldehyde in phosphate-buffered saline (PBS; Wako Pure Chemical) for approximately 20 min. After the samples were treated with 0.2% Triton X-100 for 5 min to permeabilize the cell membranes, they were blocked in an Biotin-Blocking System (DakoJapan, Kyoto, Japan) for 20 min and in TBS containing 5% BSA and 0.1% Tween 20 for 1 h at 20°C. Blocked sections were incubated in TBS + 5% BSA, 0.1% Tween 20, and rabbit anti-porcine epithelial cell adhesion molecule (EpCAM) (1:100) and sheep anti-porcine CD90, Thy-1 cell surface antigen (Thy-1) (1:25) for 24 h at 4°C, respectively. The sections were then incubated with the appropriate secondary antibody, goat anti-rabbit IgG- TRITC (1:400; Sigma–Aldrich Japan) or donkey anti-sheep IgG-Biotin (1:500; Abcam Cambridge, UK) plus ExtrAvidin-TRITC (1:180; Sigma–Aldrich Japan) for 1 h at room temperature, respectively, and anti-mouse IgG-FITC (1:320; Sigma–Aldrich Japan) for 1 h at room temperature. In addition, pCLiP were incubated with 1 mg/mL 4,6-diamidino-2-phenylindole (Dojindo Laboratories, Kumamoto, Japan) in PBS for 20 min to stain nuclei.

### Live cell counts

Viable cell counts on days 1, 3, and 7 of culture were assessed using a Cell Counting Kit-8 (WST-8; Dojindo, Tabaru, Japan). The medium in each dish was replaced with the measurement solution (medium: CCK-8 solution = 10:1), followed by incubation in a $CO_2$ incubator for 60 min. Absorbance was measured using a Multiskan FC microplate reader (Thermo Fisher Scientific, Tokyo, Japan). Absorbance was proportional to the number of viable cells.

### Flow cytometry

As markers of liver progenitor cells, we used anti-EpCAM and anti- trophoblast cell surface antigen 2 (TROP2) in flow cytometry. pCLiP were isolated after treatment with TrypLE Express. Next, isolated pCLiP were washed with D-PBS (Wako Pure Chemical) and blocked with 1% BSA in PBS for 1 h. After blocking, the plate was washed with PBS and incubated with anti-pig EpCAM (1:200; Abcam, Cambridge, UK) and TROP2 (1:1000; Novus Biologicals, Centennial, CO, USA) antibodies for approximately 2 h.

### Hepatocyte induction assay

The hepatocyte induction assay was performed using a small-molecule cocktail protocol described previously. The harvested cells were cultured on collagen-I-coated plates at a density of $5.0 \times 10^4$ cells/cm$^2$ in STIM medium. After reaching 40–60% confluence, the medium was replaced with hepatic induction medium (HIM) for 5 days. The HIM was advanced F12 basal medium supplemented with five small molecules, FH1 (15 μM), FPH1 (15 μM), A-83-01 (0.5 μM), dexamethasone (100 nM), and hydrocortisone (10 μM). Then medium was replaced HIM with a mixture of 2% Aterocollagen (Koken) for another 2 days, followed by the above-mentioned medium containing 20 μM Forskolin (Fsk, Sigma; F3917) for another 2 days.

### Cytochrome P450 CYP3A4 activity

We evaluated cytochrome P450 CYP3A4 activity to assess the metabolic function of CLiP. To measure CYP3A4 production, the cells in each group, i.e., the healthy pCLiP group and disease-derived pCLiP group (liver disease group), were replaced with 1 mL of fresh medium 2 days before the endpoint of the differentiation process. CYP3A4 activity was measured using P450-Glo™ CYP3A4 Assays (Promega, Madison, WI, USA) according to the manufacturer's instructions. The medium was removed at the end of differentiation. After two washes with

PBS, the cells were incubated with 200 μL of Luciferin-IPA (1.5 μM, diluted in PBS) for 1 h at 37°C. The reacted substrate was then transferred to a 96-well plate and mixed with an equal volume of luciferin detection reagent for 20 min at room temperature. The relative luminometer units were then determined using the Synergy LX Multi-Mode Reader (BioTek). Each data point was normalized to the total DNA volume.

### Evaluation of proliferative ability *in vivo*

Proliferative ability *in vivo* was evaluated based on albumin synthesis after pCLiP administration with liver regenerative stimulation. pCLiP were transplanted into SD rats (male, 10 weeks old) by portal vein branch ligation (PBL) for 70% (left lateral and median lobes) of the whole liver [15]. PBL effectively enhances cell proliferation in the non-ligated liver lobes by maintaining portal venous flow accompanied by atrophy of the deportalized liver [15]. The protocol of this experiment was approved by the Research Center for Biomedical Models and Animal Welfare, Nagasaki University Graduate School of Biomedical Sciences (approval number 2102241695–5). All rats were anesthetized with isoflurane during invasive procedures.

After $2 \times 10^7$ cells were transplanted into 30% of the total liver by transportal administration using a temporary clamp, the abdomen was closed with a thread (7–0 silk) surrounding the loosened portal vein and left in the peritoneal cavity. After 24 h, the abdomen was reopened, and the remaining thread was used to ligate 70% of the portal vein branch of the liver lobe to induce liver regeneration. Immunosuppressive drugs (tacrolimus, 0.15 mg/kg/animal) were administered once before surgery and three times a week starting 2 weeks after surgery (S2 Fig).

### Analysis of EVs and microRNAs in CLiP

To analyze EVs secreted by the cultured cells, the culture supernatant was collected on day 7 of culture and sampled using the MagCapture™ exosome Isolation Kit PS Ver.2 (Fuji Pharma Co. Ltd., Tokyo, Japan). Nanoparticle tracking analysis (Nanocyte) was performed to determine the number of particles and EVs (cluster of differentiation 9, CD9-positive), quantified by ELISA, in the pCLiP culture medium. Since the standard solution was not available for the calculation, the results were compared by measured absorbance. In addition, microRNA analyses of EVs were performed using Toray 3D-Gene.

### Statistical analysis

Data are presented as the mean ± standard error of the mean of independent experiments or the mean ± standard deviation of replicates assessed in separate culture wells. The two groups were compared using *t*-tests, unless otherwise stated. Statistical significance was set at $P < 0.05$ and $P < 0.01$.

## Results

### Comparative hepatic histology of miniature pigs with and without liver disease

Macroscopically, the healthy and liver disease groups exhibited interlobular connective tissue, suggesting more pronounced fibrosis than that in the normal human liver. Additionally, the tissue of the liver disease model appeared whiter in color than that of the healthy liver, with slightly blunter and more swollen margins and liver surfaces at the lobular level (Fig 1A). H&E staining of the healthy liver model revealed no signs of fatty liver parenchyma, as expected, and the liver disease model exhibited 5–66% fatty liver parenchyma, indicating steatosis

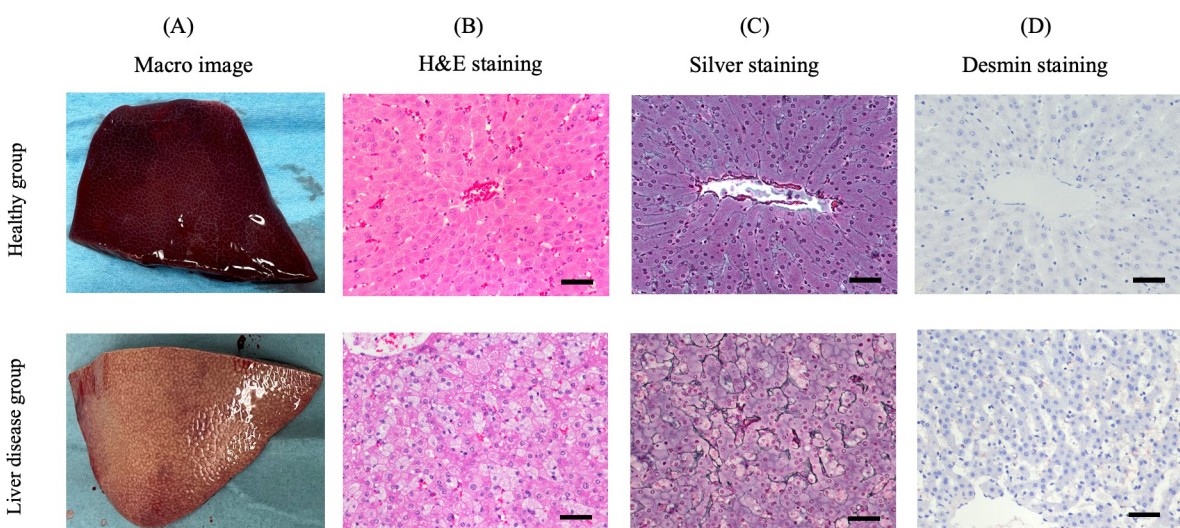

**Fig 1. Hepatic histology in healthy and liver disease model pigs.** Bars represent 50 µm.

(Fig 1B). Silver staining revealed fibrotic changes along the edge of the hepatic parenchyma in the liver disease model, with a fibrosis activity score of 1 (assigned based on a human scale) (Fig 1C). As determined by Desmin staining, the activation of liver stellate cells was observed in the disease model but not in the healthy liver model (Fig 1D). Based on these results, the liver disease model was histologically diagnosed as MASLD (Table 1 and S1 Fig).

## Characteristics of pCLiP

At 1 day of culture, the primary hepatocytes adhered to the culture plate. As they underwent division and proliferation, they began to coalesce with adjacent cells after 7 days of culture (Fig 2A). After 14 days of culture, the cells exhibited high rates of proliferation. Similar to the MH, the shape changed from a pavement-like appearance to a smaller sword-like form. Immunofluorescence staining revealed the positive expression of EpCAM and CD90 in the aggregated cells of each group (Fig 2A).

Next, cell viability was evaluated using the WST kit. Although there was no difference between the groups on day 3 of culture, at days 1 and 7 of culture, cell proliferation in the liver disease group was significantly higher than that in the healthy group (Fig 2B). A flow cytometry analysis showed that the positivity rate of EpCAM was higher in the healthy group than in the liver disease group, and the positivity rate of TROP2 was higher in the liver disease group (Fig 2C and 2D). Positivity for both antibodies indicated that the cultured cells were hepatic progenitor cells.

**Table 1. MASLD diagnostic criteria and activity score based on pathological findings (fatty deposits, fibrosis, and hepatic stellate cell activation) in healthy and liver-diseased porcine groups.**

| | | Steatosis (activity grading) | Fibrosis (staging) | Activation of hepatic stellate cells | NAS | | |
| --- | --- | --- | --- | --- | --- | --- | --- |
| | | | | | Steatosis | Lobular inflammation | Ballooning |
| Healthy group | #1 | 0 | 0 | 0 | 0 | 0 | 0 |
| | #2 | 0 | 0 | 0 | 0 | 0 | 0 |
| Liver disease group | #1 | 2 | 1 | 1 | 2 | 0 | 0 |
| | #2 | 1 | 1 | 2 | 1 | 2 | 0 |
| | #3 | 1 | 0 | 1 | 1 | 1 | 0 |

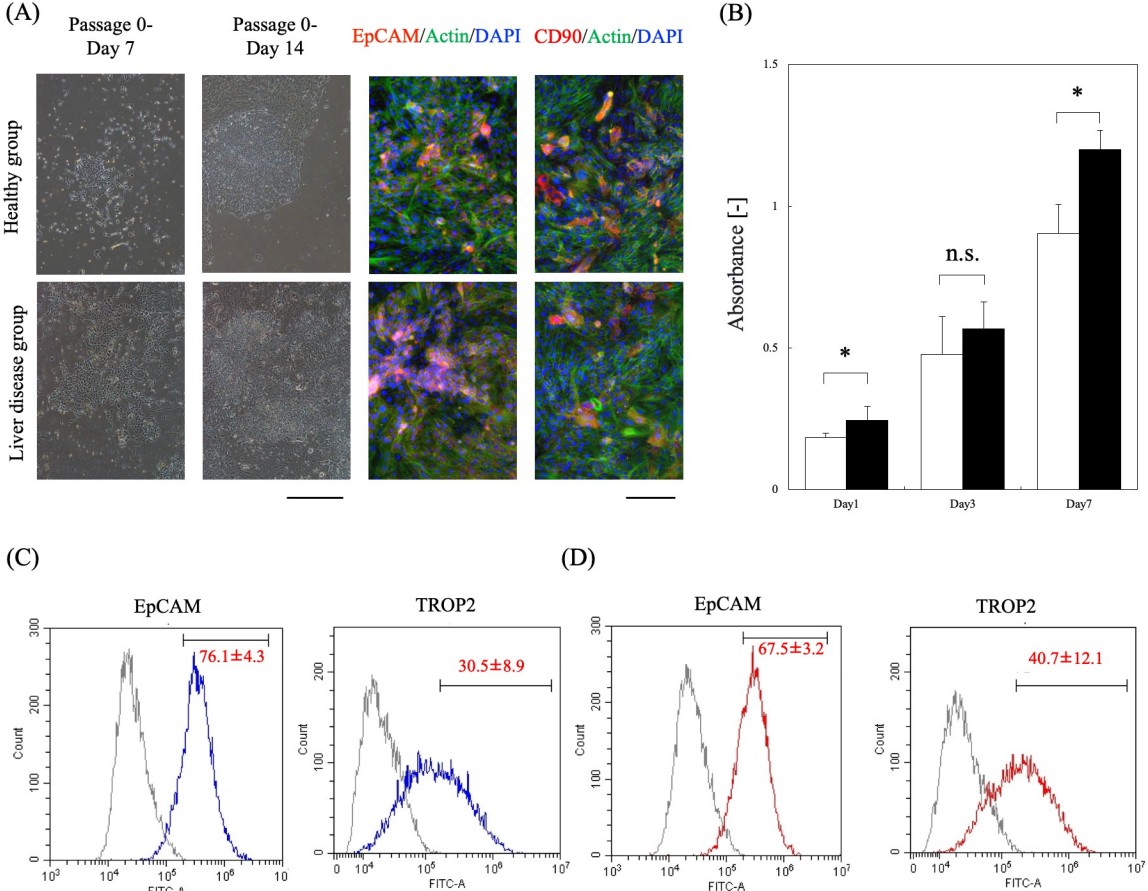

**Fig 2. Characteristics of CLiP isolated from diseased pigs.** (A) Cellular system changes during culture and immunofluorescence images at day 14 of culture; (red) hepatic progenitor cells (EpCAM and CD90), (green) cytoskeleton, (blue) nuclei. The bar for phase contrast images represents 2 mm and that for fluorescent photographic images represents 100 μm. (B) Viable cell counts during culture (n = 9 from three independent cell preparations; *t*-test, *P < 0.01). The white bar shows the healthy group. The black bar shows the liver disease group. (C) Flow cytometric analysis of pCLiP prepared from the healthy group (n = 5 from three independent cell preparations) and (D) flow cytometric analysis of pCLiP prepared from the liver disease group (n = 8 from three independent cell preparations). EpCAM, epithelial cell adhesion molecule; CD90, cluster of differentiation 90; pCLiP, porcine chemically induced liver progenitors.

## Re-differentiation of CLiP into mature cells

We further evaluated the hepatocyte re-differentiation potential of CLiP in vitro and in vivo. MH were induced in each group in vitro using a previously described method. In the healthy group without hepatocyte induction, during proliferation, small cells aggregated together, maintaining a relatively uniform appearance. However, upon the induction of hepatocyte re-differentiation, we observed re-differentiation into cells with somewhat larger nuclei and cytoplasm, resembling hepatocytes and cells with bile duct-like structures. In contrast, the non-induced liver disease group showed uniform cells that formed aggregates during proliferation. However, with hepatocyte re-differentiation induction, liquid pools were observed within the cell masses. Although not as numerous as in the healthy group, some of the liver disease groups also formed continuous lumen-like structures in these pools (Fig 3A). Regarding re-differentiation potential, an increase in CYP3A4 activity was observed for MH induced from healthy CLiP but not in those from disease-derived CLiP (Fig 3B).

Next, pCLiP from each group were transplanted into a PBL model in which liver regeneration was induced and hepatocyte re-differentiation potential was evaluated in vivo. Cells

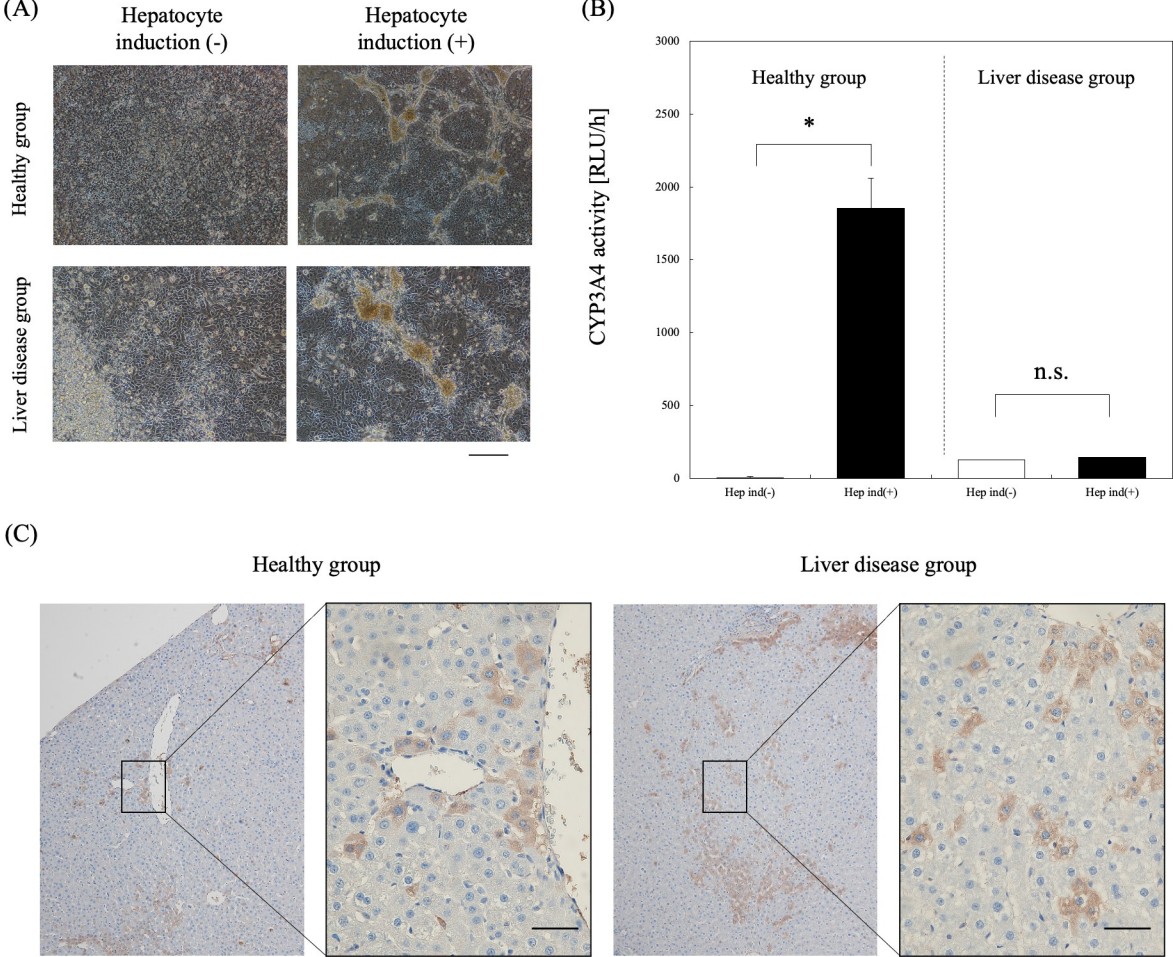

**Fig 3. Mature hepatocyte re-differentiation potential of pCLiP in vitro and in vivo.** (A) Cell morphology under various conditions at day 14 of re-differentiation induction to MH. Bars represent 200 μm. (B) CYP3A4 activity of MH at day 14 of re-differentiation (n = 6 from two independent cell preparations; *t*-test, *$P < 0.05$). (C) In vivo MH differentiation potential of pCLiP (albumin-positive) at 14 days after transplantation in the PBL model. Bars represent 500 μm.

positive for albumin, a mature hepatocyte marker, were identified in the PBL models 2 weeks after the transplantation of each CLiP (Fig 3C). This confirms that the cultured cells of each group have the ability to redifferentiate from precursor cells into mature cells in vivo.

## Analyses of EVs and microRNAs in CLiP

The quantity of nanoparticles in the culture medium was measured using a nanoparticle tracking analysis; however, no significant differences in particle quantity or cell counts were observed between the two groups (Fig 4A). However, comparing the optical density of the EVs marker CD9 in the medium was determined using a MagCapture™ exosome Isolation Kit PS Ver.2. Lower expression levels were observed in the liver disease group than in the healthy group, suggesting that there were fewer EVs in pCLiP in the MASLD group (Fig 4B). Based on a cluster analysis, among 2632 microRNAs detected, 157 microRNAs showed differences (i.e., up- or downregulation) of more than two-fold between the healthy and liver disease groups (Fig 4C). Four microRNAs related to fatty liver or liver fibrosis [16,17] did not differ

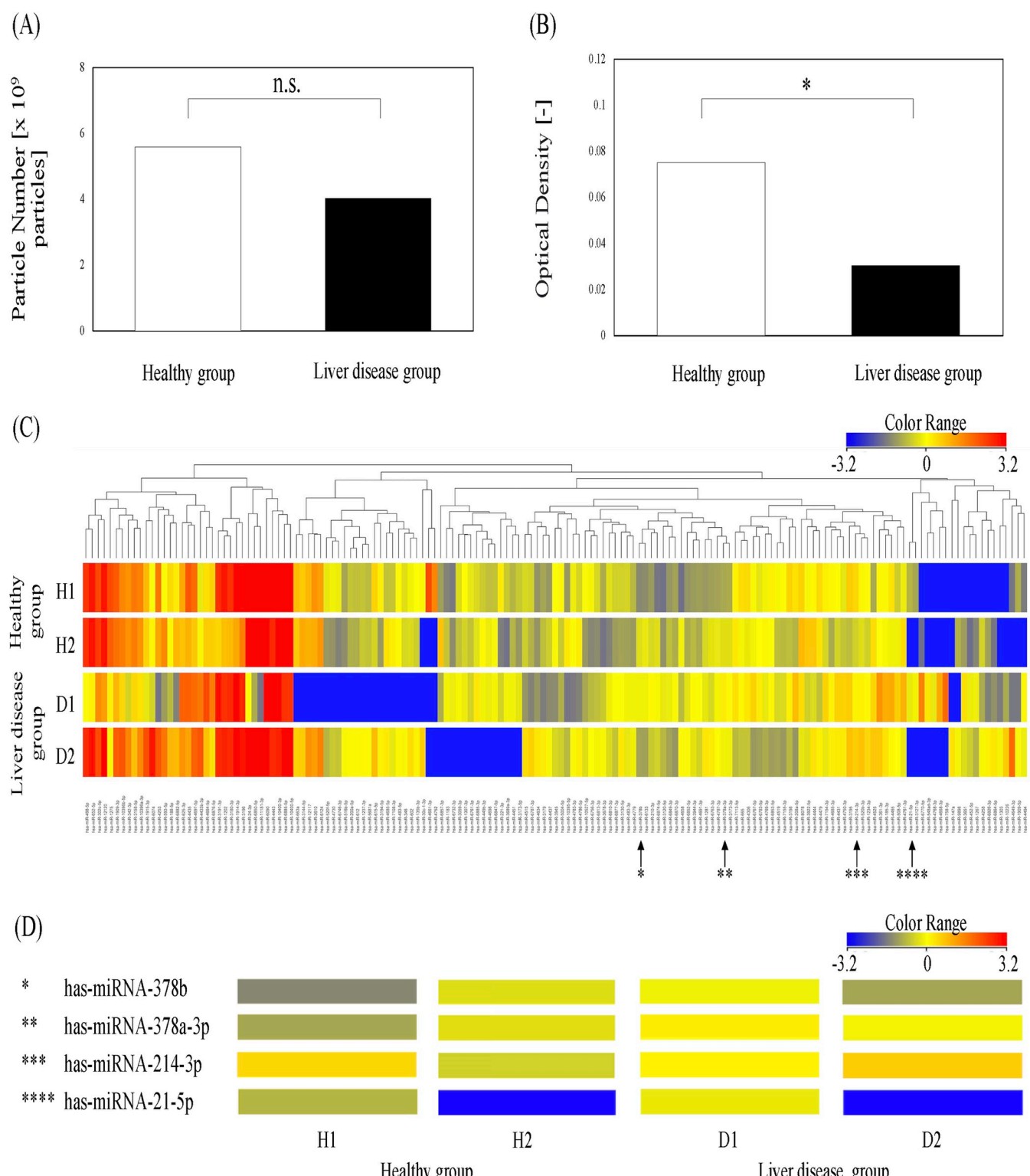

**Fig 4. Extracellular vesicles (EVs) and microRNAs in CLiP.** (A, B) Number of nanoparticles in the culture supernatant and EVs secretory activity (n = 3 from two independent cell preparations; *t*-test, *P < 0.05). (C) Heatmap of microRNA differences in a cluster analysis (n = 2 from two independent cell preparations). (D) Four microRNAs related to fatty liver or liver fibrosis were detected but did not show significant differences between the healthy and liver disease groups.

significantly between groups (Fig 4D, S1 Table). In conclusion, the cluster analysis did not reveal differences in microRNAs involved in fatty liver or liver fibrosis.

## Discussion

This study provides the first characterization of pCLiP in a pig model of diet-induced steatotic liver, reproducing MASLD. Although there were slightly fewer EVs CLiP in the disease group than in the healthy liver, the *in vivo* proliferative capacity was rather high.

We previously reported the successful production of CLiP from cirrhotic livers in humans [14]. However, the properties of CLiP depend on clinical and demographic factors [18]. Therefore, in this study, we created a dietary liver disease model and evaluated the characteristics and performance of CLiP in livers of different backgrounds.

CLiP of both groups expressed hepatic progenitor cell markers. Using flow cytometry, the EpCAM positivity rate was higher in the healthy group and the TROP2 positivity rate was higher in the liver disease group. Cell proliferation was higher for disease-derived CLiP than in healthy CLiP. Notably, TROP-2 and EpCAM were used as markers of liver progenitor cells; however, different expressions in liver progenitor cells, depending on the backgrounds of the liver, have been reported in animal models. When the liver disease progresses and fibrosis develops, TROP-2 is expressed in cholangiocytes and in activating and proliferating hepatic progenitor cells [19]. However, the cell from the healthy liver maintains a higher capacity for regeneration with relatively higher positive expression of EpCAM than the diseased liver. The phenomenon in this study may be due to the mechanism mentioned above.

Although the liver has excellent regenerative ability, Pu et al. showed that hepatic progenitor cell characteristics and liver regeneration depend on the disease [20], and Lee et al. demonstrated that mesenchymal stem cells from different sources exhibit different dedifferentiation and liver regeneration capabilities [21]. Furthermore, Kang et al. showed that differences in patient background result in variation in cellular responsiveness, such as drug sensitivity, in cancer cell assays using clinical specimens [22]. Disease-derived CLiP, with the general characteristics of hepatic progenitor cells, consistent with previous reports [13,14], showed different marker expression patterns and proliferative capacities owing to their source, i.e., MASLD with inflammation and fibrosis (Table 1 and S1 Fig). In the liver tissue of the MASLD model used in this study, the liver was stimulated to regenerate *in vivo*, and it could have regenerated vigorously after isolation and culture *in vitro*.

Furthermore, we compared the functions of mature liver cells that were redifferentiated in vitro. While the healthy group expressed markers of functional re-differentiated MHs, the liver disease group did not show sufficient expression (Fig 3). It has been reported that in MASLD, liver function decreases due to hepatocyte peroxidation and inflammation induced by Kupffer cells as well as apoptosis [23,24]. Hence, the reduced re-differentiation potential of disease-derived CLiP in vitro may reflect the decline in liver function. Varghese et al. reported that differentiating iPS cells from healthy and obese individuals into hepatocytes results in differences in fat accumulation, cell death, and gene expression related to liver fibrosis, which is a major characteristic of MASLD [25]. However, regarding the hepatocyte differentiation ability in vivo, no differences were observed in either cell group in PBL models that stimulated liver regeneration [15], confirming differentiation into mature liver cells. It has been reported that CLiP, when transplanted into the liver, stimulate liver regeneration and exhibit proliferation and re-differentiation into hepatocytes [15,26]. Although disease-derived CLiP did not form a sufficient structure for the induction of bile duct differentiation in accordance with previous studies [19], it was possible to induce bile duct differentiation by adding growth factors, such

as HGF and EGF (S3 Fig). Thus, although disease-derived CLiP have a reduced differentiation ability, they may have sufficient function as transplanted cells.

Finally, we focused on the characteristics of EVs, one of the extracellular particles, from pCLiP with different background livers [27]. The therapeutic effects of hepatic progenitor cells, including CLiP, involves the suppression of fibrosis stimulated by secreted factors as well as EVs [28–30]. Sato et al. showed that microRNA profiles and EVs characteristics differ among liver diseases [29]. Liu et al. reported the activation of stellate cells involved in liver fibrosis through an analysis of microRNAs and EVs characteristics [31]. Additionally, Takeuchi et al. reported therapeutic effects of EVs secreted by transplanted MSCs in the activation of hepatic stellate cells against liver fibrosis [9]. Similarly, we reported that transplanted CLiP reduce stellate cell activity [13]. Regarding the characteristics of EVs in vitro, there were no differences in the particle size and quantity of EVs; however, the optical density of CD9, a marker of EVs secretion, was reduced in disease-derived pCLiP (Fig 4D). Luo et al. reported high expression levels of STING1 (stimulator of interferon response cGAMP interactor 1) in patients with nonalcoholic steatohepatitis, MASLD, and in mice with diet-induced fatty liver [32]; this high expression of STING1 leads to reduced EVs secretion in normal human cells, as reported by Takahashi et al. Therefore, the reduction in EVs secretion in MASLD may reflect the pathology of the disease, that is, the characteristics of the cell source. However, since no dramatic differences in microRNA expression patterns in EVs were observed between the two groups, not only healthy CLiP but also disease-derived CLiP are candidates autologous cell sources for transplantation.

In the future, we aim to treat liver diseases through the autologous transplantation of CLiP created from patients with liver diseases. In this study, we performed laparoscopic liver resection in large animal models of liver disease, similar in size to humans, and produced pCLiP from isolated hepatocytes. By comparing the characteristics of disease-derived pCLiP and healthy pCLiP, we showed that disease-derived CLiP reflect the characteristics of the background liver, possess re-differentiation potential in vivo, and do not show differences in therapeutic effect-related microRNAs. Our future research involves the transplantation of CLiP induced from the steatotic liver into the steatotic liver to improve its function and to exert antifibrotic effects. However, a large number of cells are needed for transplantation in humans. The fact that the *in vitro* proliferative capacity of CLiPs created from diseased livers is vigorous may be an advantage when considering the future application of CLiP transplantation in humans. We are not considering applying regenerative stimuli ex vivo at this stage due to concerns about the possibility of changing the character of the cells. Stimulation factors in diseased liver are also expected to affect the *in vivo* proliferation of transplanted cells.

In this study, liver tissue was collected, stored at low temperatures, and transported, followed by cell isolation and culture. Then, the induction into CLiP was performed. Since the study did not examine the effects of preservation and transport, there may be even better conditions for better preservation methods. For clinical application, the antifibrotic effect of liver transplantation of the created CLiPs needs to be evaluated, and the evaluation of the effect of storage conditions is also a future research topic. In addition, why the regeneration of CLiPs from damaged livers was more vigorous than that of normal livers remains unclear. Notably, the efficiency of CLiP production from impaired livers may be improved if this is clarified in future studies.

In conclusion, CLiP induced in the steatotic liver are a promising source for cell therapy in patients with liver disease.

## Supporting information

**S1 Fig. Pathological findings and MASLD scores in the liver disease group.** Steatosis score 2 (A), fibrosis stage 1 (B), activation of hepatic stellate cells score 2 (C), and lobular inflammation score 1 (D). MASLD, metabolic-dysfunction-associated steatotic liver disease.
(TIF)

**S2 Fig. Portal vein administration of pCLiP to PBL models and stimulation of liver regeneration.** pCLiP, porcine chemically induced liver progenitors; PBL, portal vein branch ligation.
(TIF)

**S3 Fig. PCLiP-derived biliary network in bile duct induction.** (A) Cell morphology after induction and evaluation of bile duct transporter activity with and without transporter inhibitor (verapamil [VRP]) using rhodamine 123 stainings (green and/or red). Bars represent 50 μm. (B) Immunofluorescence staining of the bile duct network structure of disease-derived pCLiP. Red: Complete ectoderm marker (EpCAM) or cholangiocyte marker (CK7); blue: Cell nuclei. Bars represent 50 μm. pCLiP, porcine chemically induced liver progenitors; EpCAM, epithelial cell adhesion molecule.
(TIF)

**S1 Table. Normalized intensity values for microRNAs selected for clustering.**
(DOCX)

## Acknowledgments

The pathological results of the porcine tissues were evaluated by the Laboratory of Veterinary Pathology at Azabu University (and by Dr Yoko Kakinuma, Takanori Shiga, Naoyuki Aihara et al).

## Author Contributions

**Conceptualization:** Masayuki Fukumoto, Akihiko Soyama, Takahiro Ochiya, Susumu Eguchi.

**Data curation:** Masayuki Fukumoto, Takanobu Hara, Yasuhiro Maruya, Peilin Li, Hajime Matsushima, Kazushige Migita, Takahiro Enjoji, Hanako Tetsuo, Takuro Fujita, Mampei Yamashita, Hajime Imamura.

**Formal analysis:** Masayuki Fukumoto.

**Funding acquisition:** Susumu Eguchi.

**Investigation:** Masayuki Fukumoto, Daisuke Miyamoto, Takanobu Hara, Hajime Imamura.

**Methodology:** Masayuki Fukumoto, Daisuke Miyamoto, Susumu Eguchi.

**Project administration:** Susumu Eguchi.

**Supervision:** Daisuke Miyamoto, Akihiko Soyama, Tomohiko Adachi, Kengo Kanetaka, Takahiro Ochiya, Susumu Eguchi.

**Writing – original draft:** Masayuki Fukumoto, Daisuke Miyamoto, Akihiko Soyama.

**Writing – review & editing:** Akihiko Soyama, Tomohiko Adachi, Kengo Kanetaka, Takahiro Ochiya, Susumu Eguchi.

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
