## [Decision Letter · Decision Letter 0]

5 Jul 2024

PONE-D-24-20188Characteristics of chemically induced liver progenitors derived from a pig model of metabolic dysfunction-associated steatotic liver diseasePLOS ONE

Dear Dr. Eguchi,

Thank you for submitting your manuscript to PLOS ONE. After careful consideration, we feel that it has merit but does not fully meet PLOS ONE’s publication criteria as it currently stands. Therefore, we invite you to submit a revised version of the manuscript that addresses the points raised during the review process.

Please address the reviewers' concerns listed below. In particular, it will be helpful to measure the protein level in exosomes. Please also provide necessary clarification and supplementation of information.

We look forward to receiving your revised manuscript.

Kind regards,

Hon Fai Chan, PhD

Academic Editor

PLOS ONE

Journal Requirements:

2. To comply with PLOS ONE submissions requirements, in your Methods section, please provide additional information regarding the experiments involving animals and ensure you have included details on methods of anesthesia and/or analgesia, and efforts to alleviate suffering."

MVR (Straive) 20 May 2024: Please send back the ‘Tables Not In Manuscript File (in the case of file type PDF)’ send back text:

‘Please include your tables as part of your main manuscript and remove the individual files. Please note that supplementary tables (should remain/ be uploaded) as separate "supporting information" files.

4. Thank you for stating the following financial disclosure: "the Japan Agency for Medical

Research and Development (AMED)"

Additional Editor Comments:

Please address the reviewers' concerns listed below. In particular, it will be helpful to measure the protein level in exosomes. Please also provide necessary clarification and supplementation of information.

Reviewers' comments:

Reviewer's Responses to Questions

**Comments to the Author**

1. Is the manuscript technically sound, and do the data support the conclusions?

Reviewer #1: Yes

Reviewer #2: Yes

2. Has the statistical analysis been performed appropriately and rigorously? 

Reviewer #1: Yes

Reviewer #2: Yes

3. Have the authors made all data underlying the findings in their manuscript fully available?

Reviewer #1: Yes

Reviewer #2: Yes

4. Is the manuscript presented in an intelligible fashion and written in standard English?

Reviewer #1: Yes

Reviewer #2: Yes

5. Review Comments to the Author

Reviewer #1: In this study, the authors generated the CLiP derived from steatotic livers using a pig model, and evaluated their characteristics including liver-specific function, proliferative capacity in vivo, and exosome production. CLiP may be a promising source for cell therapy in patients with liver disease. Some questions should be impbroved in this study:

1. For the experiment of evaluation of proliferative ability in vivo, why do the authours chose the portal vein branch ligation (PBL) model in SD?

2. What is the proliferation rate of the pCLiP from the diseased liver? As for the futrue tansplantaion application in human, a large number of cells are needed. Do the authors have some methods for quickly enlarge the CLiP?

3. EpCAM and TROP2 are two hepatic progenitor cell markers, why the positivity rate of EpCAM was higher in the healthy group, while the positivity rate of TROP2 was higher in the liver disease group?

4. Some shortcomings should be discussed.

Reviewer #2: Title: Characteristics of chemically induced liver progenitors derived from a pig model of

metabolic dysfunction-associated steatotic liver disease

The authors reported that CLiP induced in the steatotic liver has the higher potential in the proliferative capacity than one in the healthy liver and the CLiP in both groups has the high cell viability ant the ability to differentiate into the albumin-positive cells. In addition, Exosome counts were lower in disease-derived CLiP than in the normal group; however, there were no differences in miRNA expression within exosomes. Finally, the authors mentioned that CLiP induced in the steatotic liver are a promising source for cell therapy in patients with liver disease. Although this study seemed very interesting, this manuscript needs some improvement in organization and a little more thought needs to be given to it. In addition, there are some questions in the manuscript. So, they should answer some question below:

Major revision

1: The authors explained that the mRNA levels of EVs secretory activity was proved. However, they did not described the protein level. Actually, I think that the authors need to check the protein level to establish if mRNA is working or not. Please add the explanation.

2: Extracellular vesicles can be broadly classified into three types based on their intracellular production mechanisms: exosomes, microvesicles, and apoptotic bodies. However, I felt that the authors used EVs and exosomes as if they are equivalent. I think the exosome and EVs experiments are important. Please add the explanation about them.

3: Why was the proliferative capacity in CLiP in the disease liver the higher that one in the normal liver? Liver disease itself already stimulated the capacity? Please add the explanation.

Minor revision

1: In the introduction, the authors explained MASLD in the last sentences. The order of the sentences seems odd; wouldn't MASLD's explanation be better at the beginning?

2: Why was CYP3A4 evaluated? The authors have not stated why it was evaluated, so please provide an explanation.

3: Why was CD133 positivity rate was measured in order to evaluate EpCAM and TROP2 in Flow cytometry? Please add the explanation.

4: Too many abbreviations and terms are used that are not listed in the Material and Methods. Please explain what they are used to evaluate, such as CD90, CD9, TROP2, PBLmodel, EpCAM, etc.

5: Regarding the immunostaining, the background is too irregular, especially in Figs. 1 and 3. I would like to see them changed to clearer pictures.

6. PLOS authors have the option to publish the peer review history of their article (what does this mean?). If published, this will include your full peer review and any attached files.

Reviewer #1: No

Reviewer #2: No

---

## [Author Response · Author response to Decision Letter 0]

29 Sep 2024

Responses to reviewers’ comments

We would like to thank the reviewers for providing constructive critiques that helped us improve our manuscript. We have tried to address the issues raised and respond to all comments. The revisions are indicated in red font in the revised manuscript. Please find below a detailed, point-by-point response to the reviewers' comments. We hope that our revisions now meet the reviewers' expectations.

Reviewer #1: In this study, the authors generated the CLiP derived from steatotic livers using a pig model, and evaluated their characteristics including liver-specific function, proliferative capacity in vivo, and exosome production. CLiP may be a promising source for cell therapy in patients with liver disease. Some questions should be impbroved in this study:

1. For the experiment of evaluation of proliferative ability in vivo, why do the authours chose the portal vein branch ligation (PBL) model in SD?

Response: Thank you for the important comment. PBL reportedly effectively enhances cell proliferation in one part of the liver by maintaining portal venous flow, accompanied by apoptosis in the other part of the liver without portal flow. We conducted PBL to enhance the proliferation and differentiation of CLiPs in vivo. 

We have added the following sentences on lines 213-215.

“PBL effectively enhances cell proliferation in the non-ligated liver lobes by maintaining portal venous flow accompanied by atrophy of the deportalized liver ”

2. What is the proliferation rate of the pCLiP from the diseased liver? As for the future transplantation application in human, a large number of cells are needed. Do the authors have some methods for quickly enlarge the CLiP?

Response: Thank you for your comment. The proliferation rate of pCLiPs from liver disease was higher than that of pCLiPs from healthy livers. The WST kit was used to evaluate the cell number; the results are shown in Figure 2B.

As the reviewer mentioned, a large number of cells are needed for transplantation in humans. The vigorous proliferative capacity of CLiPs created from diseased livers may be an advantage when considering future applications of CLiP transplantation in humans. However, we are not considering applying regenerative stimuli ex vivo at this stage due to concerns about the possibility of changing the character of the cells.

We have added the following contents in the Discussion section on lines 429–435. 

“However, a large number of cells are needed for transplantation in humans. The fact that the in vitro proliferative capacity of CLiPs created from diseased livers is vigorous may be an advantage when considering the future application of CLiP transplantation in humans. We are not considering applying regenerative stimuli ex vivo at this stage due to concerns about the possibility of changing the character of the cells. Stimulation factors in diseased liver are also expected to affect the in vivo proliferation of transplanted cells.”

3. EpCAM and TROP2 are two hepatic progenitor cell markers, why the positivity rate of EpCAM was higher in the healthy group, while the positivity rate of TROP2 was higher in the liver disease group?

Response: Thank you for your comment. TROP-2 was mainly located in ductular reactions and cholangiocytes, whereas EpCAM was expressed in a large part of the intermediate hepatocyte population. TROP-2 is an epithelial marker specifically expressed in activated progenitor cells in mouse models of liver disease. When the liver disease progresses, and fibrosis develops, TROP-2 is expressed in cholangiocytes and in activating and proliferating hepatic progenitor cells.

Therefore, the diseased liver activates the regeneration of the TROP-2-positive hepatic progenitor cells compared with the healthy liver. The cells from the healthy liver have a higher capacity for regeneration and a relatively higher positive expression of EpCAM.

We have added the following contents in the Discussion section on lines 361–368.

“Notably, TROP-2 and EpCAM were used as markers of liver progenitor cells; however, different expressions in liver progenitor cells, depending on the backgrounds of the liver, have been reported in animal models. When liver disease progresses and fibrosis develops, TROP-2 is expressed in cholangiocytes and in activating and proliferating hepatic progenitor cells [19]. However, the cell from the healthy liver maintains a higher capacity for regeneration with relatively higher positive expression of EpCAM than the diseased liver. The phenomenon in this study may be due to the mechanism mentioned above.”

4. Some shortcomings should be discussed.

Response: Thank you for your comment. We have added the following contents in the Discussion section on lines 436–444.

“In this study, liver tissue was collected, stored at low temperatures, and transported, followed by cell isolation and culture. Then, the induction into CLiP was performed. Since the study did not examine the effects of preservation and transport, there may be even better conditions for better preservation methods. For clinical application, the antifibrotic effect of liver transplantation of the created CLiPs needs to be evaluated, and the evaluation of the effect of storage conditions is also a future research topic. In addition, why the regeneration of CLiPs from damaged livers was more vigorous than that of normal livers remains unclear. Notably, the efficiency of CLiP production from impaired livers may be improved if this is clarified in future studies.”

 Reviewer #2: Title: Characteristics of chemically induced liver progenitors derived from a pig model of metabolic dysfunction-associated steatotic liver disease

The authors reported that CLiP induced in the steatotic liver has the higher potential in the proliferative capacity than one in the healthy liver and the CLiP in both groups has the high cell viability ant the ability to differentiate into the albumin-positive cells. In addition, Exosome counts were lower in disease-derived CLiP than in the normal group; however, there were no differences in miRNA expression within exosomes. Finally, the authors mentioned that CLiP induced in the steatotic liver are a promising source for cell therapy in patients with liver disease. Although this study seemed very interesting, this manuscript needs some improvement in organization and a little more thought needs to be given to it. In addition, there are some questions in the manuscript. So, they should answer some question below:

Major revision

1: The authors explained that the mRNA levels of EVs secretory activity was proved. However, they did not describe the protein level. Actually, I think that the authors need to check the protein level to establish if mRNA is working or not. Please add the explanation. 

Response: Thank you for the comment. In this study, we did not evaluate mRNA levels of EVs but the micro RNA expression pattern of EVs by microarray. We modified the expression in the manuscript from miRNA to microRNA for clearer recognition.

2: Extracellular vesicles can be broadly classified into three types based on their intracellular production mechanisms: exosomes, microvesicles, and apoptotic bodies. However, I felt that the authors used EVs and exosomes as if they are equivalent. I think the exosome and EVs experiments are important. Please add the explanation about them.

Response: Thank you very much for your very vital remarks. In this study, we used the MagCapture Exosome Isolation Kit PS to evaluate EVs. However, as the reviewer pointed out, we did not differentiate between the exosomes, microvesicles, and apoptotic bodies. In the manuscript, we used an inappropriate mixed expression of extracellular vesicles and exosomes. Therefore, we have unified our notation to extracellular vesicles (EVs).

3: Why was the proliferative capacity in CLiP in the disease liver the higher that one in the normal liver? Liver disease itself already stimulated the capacity? Please add the explanation. 

Response: Thank you for asking this important question. As the reviewer commented, various cell proliferation stimulating and tissue repair factors are released from the diseased liver, and we expect that these factors will promote the proliferation and differentiation of CLiP. We cited the previous reports on the different proliferative potential of cells depending on the background liver as follows in the Discussion section, on lines 361–368.

Additionally, as per the reviewer's comment, we noted in the Discussion section that the steatohepatitis tissue used in this study originally had a regenerative stimulus, and thus, the cells may have regenerated vigorously even after isolation and culture. We added the following sentence in the Discussion section on lines 378–380.

“In the liver tissue of the MASLD model used in this study, the liver was stimulated to regenerate in vivo, and it could have regenerated vigorously after isolation and culture in vitro.”

In this study, we did not elucidate the mechanisms for the above content, and we consider this an issue for future studies. We have added a note to that effect in the Discussion section as a limitation of this study on lines 441–444.

“In addition, why the regeneration of CLiPs from damaged livers was more vigorous than that of normal livers remains unclear. Notably, the efficiency of CLiP production from impaired livers may be improved if this is clarified in future studies.”

Minor revision

1: In the introduction, the authors explained MASLD in the last sentences. The order of the sentences seems odd; wouldn't MASLD's explanation be better at the beginning?

Response: Thank you for the comment. We explained MASLD as a common indication of liver transplantation at the beginning of the introduction on lines 48-50. We have also added relevant explanations about our previous work in the Introduction section on lines 71–74.

2: Why was CYP3A4 evaluated? The authors have not stated why it was evaluated, so please provide an explanation.

Response: Thank you for the comment. We evaluated CYP3A4 as a marker of metabolic function of CLiPs. 

We added the following sentence in lines 196–197. 

“We evaluated cytochrome P450 CYP3A4 activity to assess the metabolic function of CLiP.”

3: Why was CD133 positivity rate was measured in order to evaluate EpCAM and TROP2 in Flow cytometry? Please add the explanation. 

Response: Thank you for the comment. CD133 is used as one of the markers to characterize mouse, rat, and human CLiPs. Initially, we had considered evaluating it in porcine. However, we were unable to obtain appropriate antibodies for pig evaluation and consequently did not perform the study. The description of the planning phase was retained in the manuscript for submission. We apologize for the inconvenience. In this revision, we have removed this information.

4: Too many abbreviations and terms are used that are not listed in the Material and Methods. Please explain what they are used to evaluate, such as CD90, CD9, TROP2, PBL model, EpCAM, etc.

Response: Thank you for the comment. We have modified the Material and Methods section. We have also made an abbreviation list and attached the list to the submitted files.

5: Regarding the immunostaining, the background is too irregular, especially in Figs. 1 and 3. I would like to see them changed to clearer pictures.

Response: Thank you for the comment. We have changed the previous pictures to clearer pictures with a higher resolution.

---

## [Decision Letter · Decision Letter 1]

23 Oct 2024

Characteristics of chemically induced liver progenitors derived from a pig model of metabolic dysfunction-associated steatotic liver disease

PONE-D-24-20188R1

Dear Dr. Eguchi,

We’re pleased to inform you that your manuscript has been judged scientifically suitable for publication and will be formally accepted for publication once it meets all outstanding technical requirements.

Kind regards,

Hon Fai Chan, PhD

Academic Editor

PLOS ONE

Additional Editor Comments (optional):

Reviewers' comments:

Reviewer's Responses to Questions

**Comments to the Author**

1. If the authors have adequately addressed your comments raised in a previous round of review and you feel that this manuscript is now acceptable for publication, you may indicate that here to bypass the “Comments to the Author” section, enter your conflict of interest statement in the “Confidential to Editor” section, and submit your "Accept" recommendation.

Reviewer #1: All comments have been addressed

Reviewer #2: All comments have been addressed

2. Is the manuscript technically sound, and do the data support the conclusions?

Reviewer #1: Yes

Reviewer #2: Yes

3. Has the statistical analysis been performed appropriately and rigorously? 

Reviewer #1: Yes

Reviewer #2: Yes

4. Have the authors made all data underlying the findings in their manuscript fully available?

Reviewer #1: Yes

Reviewer #2: Yes

5. Is the manuscript presented in an intelligible fashion and written in standard English?

Reviewer #1: Yes

Reviewer #2: Yes

6. Review Comments to the Author

Reviewer #1: The authors have adequately addressed the comments.

Do the authors have tried to re-differentiate the pCLiP into mature bile duct cells?

Reviewer #2: The author has responded to the questions for major and minor revision accurately, so I will accept the paper for this journal.

7. PLOS authors have the option to publish the peer review history of their article (what does this mean?). If published, this will include your full peer review and any attached files.

Reviewer #1: No

Reviewer #2: No

---

## [Editor Report · Acceptance letter]

28 Oct 2024

PONE-D-24-20188R1 

PLOS ONE

Dear Dr. Eguchi, 

I'm pleased to inform you that your manuscript has been deemed suitable for publication in PLOS ONE. Congratulations! Your manuscript is now being handed over to our production team.

Kind regards, 

on behalf of

Professor Hon Fai Chan 

Academic Editor

PLOS ONE